# Efficiency Improvement for Wireless Power Transfer System via a Nonlinear Resistance Matching Network

**Haonan Yang, Chengming Wu * and Tie Chen**

College of Electrical Engineering and New Energy, China Three Gorges University, Yichang 443002, China
* Correspondence: 13972603603@ctgu.edu.cn

**Abstract:** The primary control is widely adopted to obtain a constant voltage output under a wider load range. However, for traditional full-bridge inverters under phase-shift control, due to the loss of soft switching, the system transmission efficiency will decrease rapidly. This problem can be improved by using a half-bridge inverter; however, the power transferred to receiving devices utilizing a half-bridge inverter is inadequate under a small dc load value. To solve these urge issues, a resistance matching network (RMN), constructed by a resonant inductor and capacitor, is added before the full bridge rectifier part of this paper. Owing to the nonlinear resistance conversion characteristics of the proposed RMN structure, the variation range of the equivalent AC impedance before the rectifier end can be compressed into an optimum range to enhance the poor transmission efficiency during the coils part. Simultaneously, when the DC load is small, the power capability of the whole system can be enhanced by increasing the equivalent ac load value; as a result, the insufficient power capability of the half-bridge inverter can be relieved. Finally, a 36 V output WPT system based on the proposed RMN and phase shift control is constructed. Additionally, the experimental results prove the feasibility of the theoretical analysis results.

**Keywords:** wireless power transfer (WPT); phase shift (PS) control; resistance matching network (RMN); efficiency improvement; power capability enhancement





## 1. Introduction

Nowadays, more and more studies on wireless power transfer (WPT) technology have been conducted due to its safety, flexibility, and convenience. Because the wire connection is avoided, it has been adopted in many different applications, such as implantable devices [1], consumer electronics [2], and electrical vehicles (EVs) [3].

Among them, wireless charging for EVs (Figure 1) is considered the focus technology in the future. The DC bus voltage of EVs needs to be kept at a constant value under any conditions [4]. Normally, the realization of constant voltage (CV) can be implemented by the following methods. (1) Pulse frequency modulation (PFM): a stable charging voltage can be proposed by adjusting the operation frequency automatically with the variation of wide range mutual inductance [5]. However, the wide range of frequency changes will deprive the system of reliability and stability due to the emergence of compensation network detuning. (2) The addition of an additional dc/dc converter: an extra dc/dc converter with pulse width modulation (PWM) is adopted at the transmitter side to regulate the output voltage [6]. Nevertheless, the power density of the whole system will decrease due to the additional dc/dc converters, not to mention extra power losses and increased costs. (3) Compensation network [7,8]: a proposed compensation topology, named S/CLC, provides the advantages of constant voltage output and realization of nearly zero phase angle (ZPA) and zero voltage switching (ZVS) when the load varies [7]. However, the output voltage is fixed and cannot be regulated flexibly. Moreover, the output voltage value will be affected by the coupling coefficient. (4) Variable parameters adjustment [9,10]: a variable inductance is adopted on the primary side to maintain a constant voltage output [9]. Nevertheless, the

control of variable inductance requires an extra buck converter and corresponding feedback circuits. (5) Phase shift control [11,12]: due to the application of phase shift control, the four power switches of the full-bridge (FB) inverter all work under a hard-switching case, which increases the whole system loss. The half-bridge (HB) inverter with only two power switches can increase the switching loss. However, the power output capability (POC) of the half-bridge inverter is not enough, especially under light-load conditions.

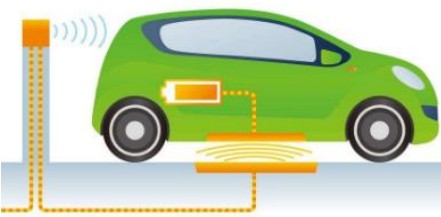

**Figure 1.** Wireless charging technology for EVs.

In addition, for the coupling efficiency of any WPT systems with fixed circuit parameters, a proper load $R_{opt}$ exists to obtain the maximum coupling efficiency [13]. To realize the maximum efficiency transmission during the whole load variation range, lots of related studies have been conducted. An additional dc/dc converter, cascaded between the rectifier and dc load, is adopted to realize the conversion of impedance, then maximum coupling efficiency can be kept during the load variation range [14]. However, a cascaded dc/dc converter will decrease the system efficiency due to the hard switching operation state. In addition, extra installment space and components cost brought by the dc/dc converter must be considered as well. Different impedance-matching networks, constructed by multiple inductances or capacitors, are inserted between the rectifier part and receiving coil to achieve impedance conversion [15,16]. Nevertheless, apart from the extra component space and cost, additional imaginary impedance generated during the conversion process is a worse issue. By comparison, by regulating the conduction duty cycle of the semi-active rectifier, maximum coupling efficiency can be satisfied due to the realization of optimal equivalent impedance [17–19]. However, the complexity of the whole regulation process will increase due to the desirability of crossing zero detection. In addition, some other research is mainly focused on mutual inductance variation [20] or low-power occasions [21], which does not work well for EVs.

To improve those issues, a resistance compression network (RMN) is installed in front of the rectifier part of this paper. When the load value is small, by designing the RMN parameters properly, the input impedance of RMN can be converted to increase the power transmission capability of the proposed WPT system. Similarly, after a proper choice of RMN parameters, the dc load with a wide variation range can be converted around optimal ac load for system efficiency improvement. Then, a WPT system, utilizing a half-bridge inverter with a phase shift control method on the primary side and an RMN on the secondary side, is proposed. In short, the insufficient power transmission capability of the half-bridge inverter can be eliminated and the dc/dc transmission efficiency of the proposed WPT system will be greatly enhanced by the application of the adopted RMN structure.

## 2. Theoretical Analysis

In this section, a WPT system with S/S (the first S means the series compensation on the primary, and the second S represents the series compensation on the secondary side) compensation network is taken as an example for characteristics analysis.

### 2.1. Topology Simplification

A WPT system utilizing an S/S topology is shown in Figure 2, where the $v_p$ and $v_s$ represent the ac input voltage source and ac output voltage, respectively. The resonant

network on the transferring side is composed of transferring coil $L_p$, compensation capacitor $C_p$, and the equivalent series resistance (ESR) $R_p$ of the whole branch. On the secondary side, receiving coil $L_s$, resonant capacitors $C_s$ and $R_s$ form the secondary compensation network. $R_{eq}$ is the equivalent input impedance of the whole rectifier part. $M$ is determined by the distance between the transferring and receiving coils, and it will affect the energy received on the secondary side. $i_p$ and $i_s$ represent the loop currents in the primary and secondary resonant network, respectively.

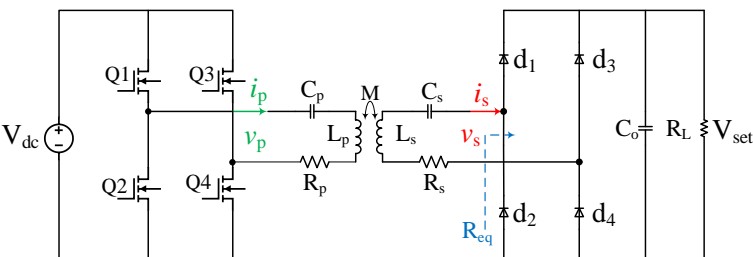

**Figure 2.** A common WPT system utilizing an S/S compensation network.

The key waveforms of S/S compensation WPT system utilizing a PS control are drawn in Figure 3. *a* is the shift phase angle, and $d_1$ can be seen as the pulse width of the output voltage of the inverter part.

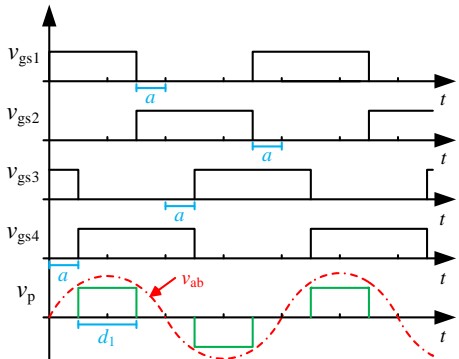

**Figure 3.** Main waveforms for WPT system with a PS control.

Since the amplitude of the fundamental wave in the square wave is much higher than that of other frequency harmonics, the fundamental component approximation (FHA) method is adopted to simplify the calculation process in this paper. The fundamental component magnitude of the high-frequency inverter (HFI) output voltage $v_p$ with duty cycle $d_1$ can be expressed as the following equation:

$$v_p = K_p V_{dc} \sin(\frac{d_1}{2}) \tag{1}$$

Normally, the expression of $K_p$ for the full bridge (FB) and half-bridge (HB) inverter can be given as follows:

$$\begin{aligned} K_{p-FB} &= \frac{2\sqrt{2}}{\pi} \\ K_{p-HB} &= \frac{\sqrt{2}}{\pi} \end{aligned} \tag{2}$$

In addition, the relationship between the input resistance of the full-bridge rectifier and load $R_L$ is given by

$$R_{eq} = \frac{8R_L}{\pi^2} \tag{3}$$

## 2.2. Efficiency Analysis

### 2.2.1. Coupling Efficiency Analysis

The circuit model of the compensation network part is shown in Figure 4a, and according to the mutual inductance equivalence theory, the equivalent circuit is depicted in Figure 4b.

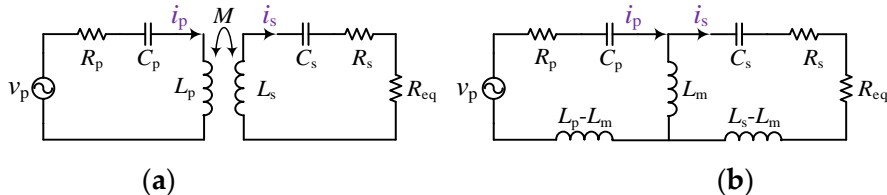

**Figure 4.** S/S topology circuit. (**a**) Primitive circuit. (**b**) Equivalent circuit.

The working angular frequency is set as $\omega$, based on the mutual inductance theory and Kirchhoff's voltage law (KVL). The expression of coupling efficiency $\eta$ can be given as

$$\eta = \frac{R_{eq}}{R_{eq} + R_s} \frac{\text{Re}[Z_r]}{\text{Re}[Z_r] + R_p} \tag{4}$$

where the reflected impedance $Z_{in}$, the primary impedance $Z_p$, and the secondary impedance $Z_s$ are defined as

$$\begin{cases} Z_r = \frac{\omega^2 M^2}{Z_s + R_{eq} + R_s} \\ Z_p = j\omega L_p - \frac{1}{j\omega C_p} \\ Z_s = j\omega L_s - \frac{1}{j\omega C_s} \end{cases} \tag{5}$$

From Equations (4) and (5), the coupling efficiency $\eta$ is related to primary branch internal resistance $R_p$, circuit parameters in the secondary resonant network, load $R_{eq}$ and mutual inductance $M$. It is noted that $R_p$ and $R_s$ are mainly the internal resistors of transferring and receiving coil. Generally, multi-strand Litz wire can be adopted to reduce them. However, the $R_p$ and $R_s$ value is constant for a couple of designed coupling coils. Therefore, the influence comes from the other three variables ($R_{eq}$, $M$ and $Z_r$) analyzed to obtain the optimum condition for efficient power transfer during the given load range.

From Equation (5), it can be seen that the maximum transmission efficiency happens when the secondary resonant tank is in a resonant state for a specific load $R_{eq}$. Furthermore, the transmission efficiency curve against load $R_{eq}$ at different M values is drawn in Figure 5. Obviously, with the increasing of $R_{eq}$, a maximum coupling efficiency point occurs, and the corresponding load point is considered as the optimum load $R_{opt}$. In addition, the point of $R_{opt}$ will be affected by the mutual inductance $M$. With a larger M value, the value of $R_{opt}$ will increase accordingly. More importantly, once $R_{eq}$ deviates from $R_{opt}$, the coupling efficiency will fall rapidly, especially under relatively weak coupling conditions. By taking the derivative of (4) with respect to $R_{eq}$, the value of $R_{opt}$ can be deduced when other circuit parameters are given in advance, and the expression of $R_{eq-opt}$ for maximum coupling efficiency is presented as

$$R_{eq-opt} = R_s \sqrt{1 + \frac{\omega^2 M^2}{R_p R_s}} \tag{6}$$

In summary, to keep the coupling efficiency at an optimum level under the whole load variation range, the following conditions should be considered.

(a)  The working angular frequency should be equal to the inherent resonant frequency of the secondary network as

$$\omega = \frac{1}{\sqrt{L_s C_s}} \tag{7}$$

(b)　The variable load $R_{eq}$ should be matched to $R_{eq\text{-}opt}$ (or close to $R_{eq\text{-}opt}$) by impedance matching network or other methods, especially in the weak coupling occasion. In addition, it is noted that the non-resistive impedance is not suggested to be introduced during the used impedance matching methods, or the resonant condition (a) will be destroyed.

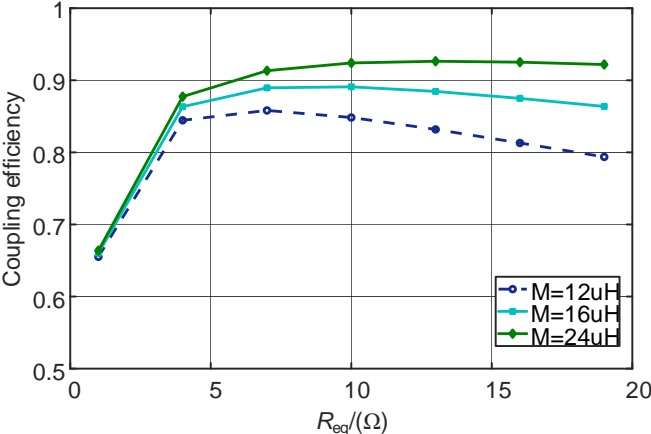

**Figure 5.** Calculated coupling efficiency curves against $R_{eq}$ at different $M$ values. (Calculation condition: $L_p$ = 162.4 uH, $L_s$ = 162.4 uH, $f$ = 85 kHz, $R_p = R_s$ = 0.5 Ω).

### 2.2.2. Efficiency Comparison between FB and HB Inverter

For an inverter utilizing a PS control method, the soft switching of all power switches during the inverter part will be lost. As a result, switching loss $P_{sw}$ dominates mostly in the inverter partial loss. A schematic diagram for the switching loss of the power switch is depicted in Figure 6. For example, when $Q_4$ is turned on, the voltage $V_{ds4}$ of $Q_4$ does not immediately drop to zero but has a fall time. During this period, the current $i_p$ and voltage $V_{ds4}$ of $Q_4$ has an overlapping area, which will cause switching loss. Additionally, the expression can be given as

$$P_{sw} = \frac{1}{6} f_{sw} t_{on} V_{dc} I_p \sin\left(\frac{\pi}{2} - \frac{\pi}{2} d_1\right) \tag{8}$$

where $t_{on}$ is the conduction time of the switch and $f_{sw}$ is the switching frequency. From (8), $d_1$ should be as large as possible to reduce the switching loss.

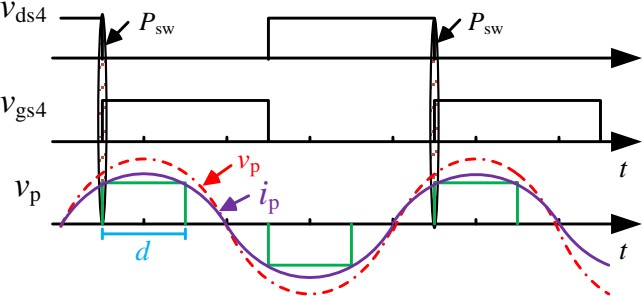

**Figure 6.** A schematic diagram for the switching loss of power switch.

To eliminate the reactive power and improve transmission efficiency, assuming the primary and secondary resonant networks are both in resonant state, which means the switching frequency should satisfy

$$\omega = \frac{1}{\sqrt{L_s C_s}} = \frac{1}{\sqrt{L_p C_p}} \tag{9}$$

From Equations (1), (2), and (4), the output power expression is deduced as

$$P_o = \frac{R_{eq} + R_s}{\omega^2 M^2 + R_{eq}(R_{eq} + R_s)} K_p^2 V_{dc}^2 \sin^2(\frac{d_1}{2}\pi) \tag{10}$$

From Equation (10), output power $P_{\mathrm{o}}$ is determined by $d_1$. It is noted that the $d_1$ will reduce accordingly with the decrease of demanded output power. However, too small $d_1$ will result in the increase of switching loss. Generally, for a well-designed system, input voltage $V_{\mathrm{dc}}$, circuit parameters, and working frequency can be considered to be constant. Only $K_{\mathrm{p}}$ can be regulated by controlling the operation mode of inverter part. From (2), $K_{\mathrm{p}}$ value of HB inverter is only half of that of FB inverter. Therefore, $d_1$ can be increased by choosing a smaller $K_{\mathrm{p}}$ when same output power is required. Further, the switching loss of inverter part can be reduced to enhance the inverter conversion efficiency.

*2.3. Power Analysis*

According to the above analysis result, WPT system utilizing an HB inverter has the ability to reduce switching loss under same output power requirement. However, its power capability should be discussed.

$$P_{out} = \frac{K_p^2 V_{dc}^2 \omega^2 M^2 R_{eq}}{(\omega^2 M^2 + R_{eq}(R_{eq} + R_s))^2} \sin^2(\frac{d_1}{2}\pi) \tag{11}$$

From Equations (4) and (10), the power received on the secondary side can be expressed as (11). Then, the relationship between output power $P_{\mathrm{out}}$ and $R_{\mathrm{eq}}$ can be depicted in Figure 7. Obviously, for S/S compensation WPT system adopting a traditional full bridge rectifier, the maximum output power will fall rapidly as $R_{\mathrm{eq}}$ decreases. As a result, there exists a problem of insufficient output power for WPT system utilizing an HB inverter under small $R_{\mathrm{eq}}$ conditions. Therefore, power enhancement by resistance matching network can be considered a possible way. In addition, some limitations should be preset as:

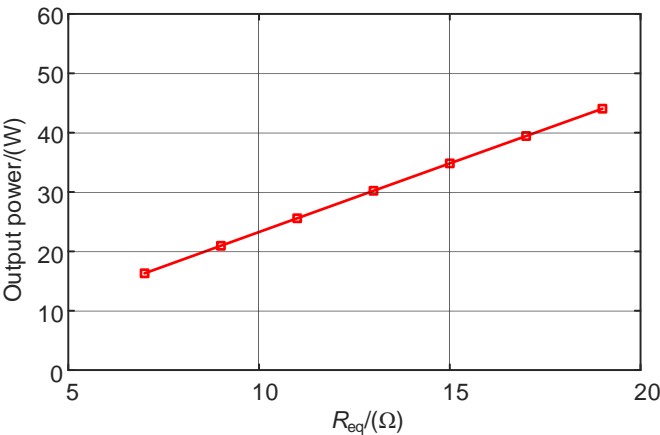

**Figure 7.** Maximum output power against $R_{\mathrm{eq}}$ for WPT system utilizing an HB inverter. (Calculation condition: $L_{\mathrm{p}}$ = 162.4 uH, $L_{\mathrm{s}}$ = 162.4 uH, $f$ = 85 kHz, $R_{\mathrm{p}}$ = $R_{\mathrm{s}}$ = 0.5 Ω, M = 30 uH).

(a) No additional imaginary impedance is generated during the conversion process;
(b) Impedance enhancement when dc load value is smaller, and impedance compression when dc load value is larger. In other words, a nonlinear conversion characteristic is required to improve the transmission performance of WPT system.

## 3. Topology Optimization

Based on the above analysis results, low coupling efficiency under a wide load range and insufficient power capability under a small load value limit the application of the WPT

system with an HB inverter. Therefore, a resistance matching network (RMN) is added to improve these issues.

To enhance the power capability, an RMN is added in front of the rectifier, which is drawn in Figure 8. It is noted that the resonant frequency of $L_r$ and $C_r$ should be equal to the switching frequency to realize the pure resistive transformation. The relationship between the input resistance of the full-bridge rectifier and load $R_L$ has been given by Equation (3). Then, equivalent resistance $R_c$ in the function of the value of the load $R_L$ for the circuit in Figure 8 is given by [22] as

$$R_c = \frac{X^2 + R_{eq}^2}{R_{eq}} \sin^2\left(\frac{\pi}{2} - \frac{1}{2}\tan\left(\frac{\omega L_r}{R_{eq}}\right)\right) \tag{12}$$

where X is defined as the impedance magnitude of used RMN elements ($L_r$ and $C_r$) under switching frequency.

$$X = \omega L_r = \frac{1}{\omega C_r} \tag{13}$$

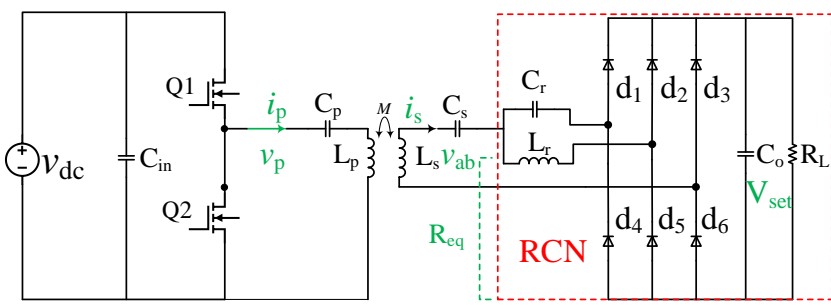

**Figure 8.** The topology of the WPT system using the proposed RMN structure.

The output power against $R_L$ at different backend structures can be depicted in Figure 9. For the S/S compensation WPT system adopting a full bridge rectifier, the maximum output power will fall rapidly as $R_L$ decreases. By comparison, the power output ability can be greatly enhanced by the implementation of the proposed RMN structure when the load value is small. As a result, the problem of insufficient output capacity for HB inverters under a small load value can be solved.

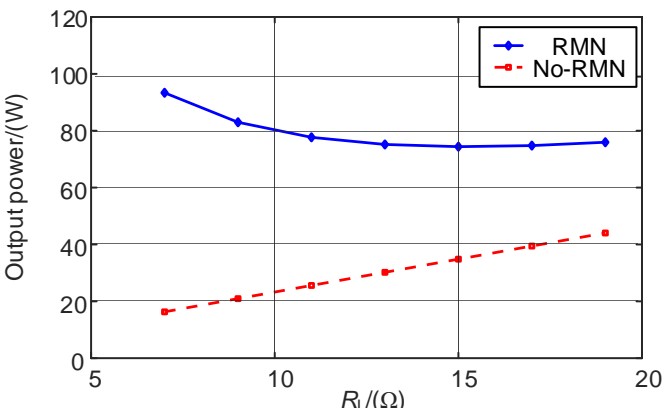

**Figure 9.** The output power against $R_L$ under different rectifier parts. (Calculation condition: $L_p$ = 162.4 uH, $L_s$ = 162.4 uH, $f$ = 85 kHz, $R_p = R_s$ = 0.5 $\Omega$, $M$ = 30 uH, $L_r$ = 25 uH).

In addition, the equivalent input impedance curves of the RMN rectifier and conventional rectifier are depicted in Figure 10. It shows that the input impedance magnitude of the RMN rectifier varies from 26.16 to 38.03 $\Omega$ when $R_L$ changes from 10 to 40 $\Omega$, while the bridge rectifier input resistance from 8.1 to 32.42 $\Omega$. The variation range of input impedance

of the RMN rectifier is only half of that of the conventional rectifier. Therefore, the input impedance magnitude of the RMN rectifier can be maintained around a relatively constant value. From Equation (4), the coupling efficiency is affected greatly by $R_L$. Therefore, the coupling efficiency against $R_L$ under different rectifier parts is also drawn in Figure 11. Since the input impedance value of the RMN rectifier is converted into near $R_{opt}$, this helps to maintain the coupling efficiency at a quite high level across a wide load variation range.

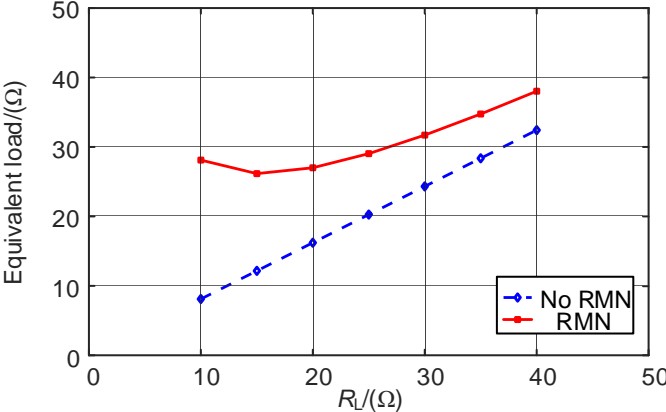

**Figure 10.** The equivalent input impedance value against $R_L$ under different rectifier parts. (Calculation condition: $f$ = 85 kHz, $L_r$ = 25 uH).

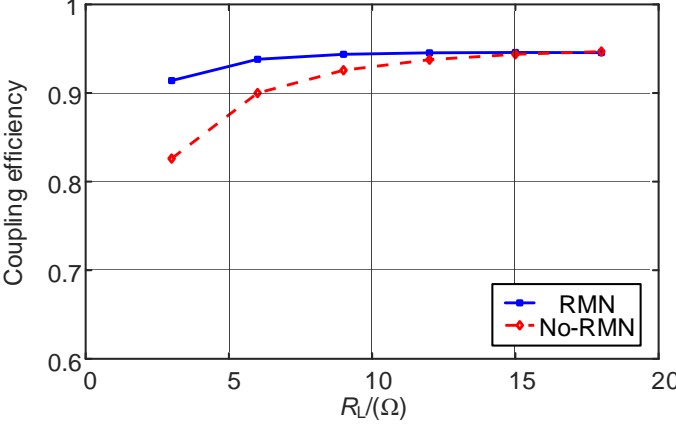

**Figure 11.** The coupling efficiency curves against $R_L$ under different rectifier parts. (Calculation condition: $L_p$ = 162.4 uH, $L_s$ = 162.4 uH, $f$ = 85 kHz, $R_p$ = $R_s$ = 0.5 Ω, $M$ = 30 uH, $L_r$ = 25 uH).

## 4. Control Strategy

The topology of the S/S compensation WPT system using the proposed RMN structure is shown in Figure 8. On the primary side, the output voltage can be satisfied to be constant by controlling the pulse width of the high-frequency inverter (HFI) output voltage, which can be adjusted in the dc/ac inverter part. The S/S compensation network is chosen as the coupling tank to realize the power transformation from the primary to the secondary side, then follows the proposed RMN structure on the secondary side. Compared with typical bridge rectifier circuits, another two diodes are needed to ensure that $L_r$ and $C_r$ are both connected in series with the load.

The common fixed-frequency control with variable $D$, being implemented in the digital signal processor (DSP), is used for the primary inverter part. The control diagram of the proposed S/S compensation WPT system is demonstrated in Figure 12. During the whole operation process, the primary resonant current $i_p$ is detected constantly. Initially, the $D$ is set as 0.5 in advance, then combined with the measured $i_p$ value, the input impedance $R_c$ of RMN can be calculated from Equation (12). Furthermore, the required phase shift angle to

realize the preset output voltage can be obtained from Equation (11). Finally, corresponding signals for $S_1$ and $S_2$ can be generated in the controller to drive the HB inverter. In addition, with a driving circuit isolated by an optocoupler, the safety issues of the control circuit can be guaranteed.

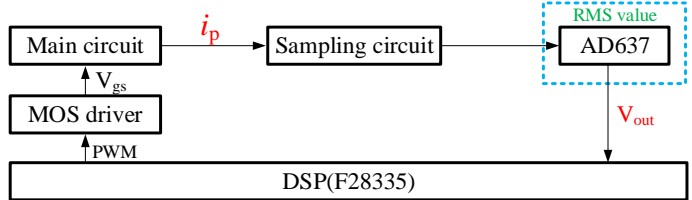

**Figure 12.** The control frame of the proposed WPT system.

## 5. Experimental Performance

### 5.1. Experimental Platform

In order to verify the practicability of the above theoretical analysis results, an S/S compensation WPT system adopting the proposed RMN structure with a 36 V output voltage was built in this section. Corresponding circuit parameters have been given in Table 1. The experimental platform is shown in Figure 13. The HB inverter is constructed by two Si MOSFETs (IRFP4227) with a lower conduction resistor. Polypropylene film capacitors are chosen to resonate with the self-inductance of coupling coils due to their better performance under high-frequency occasions. On the secondary side, six Schottky diodes PSM20U200GS are used to construct the uncontrolled RMN rectifier due to its lower forward conduction voltage drop at rated power. As for the control part, the AMC1301 produced by Texas Instruments (TI) is adopted to amplify the ac voltage signal converted by a sampling resistance. Then, an AD637 follows to obtain the RMS value of the output signal from the AM1301. Finally, the controller adopts DSP (TMS320F28335) produced by TI to process data and output corresponding control signals.

**Table 1.** Specific components of system.

| $L_p$ | $C_p$ | $L_s$ | $C_s$ | $C_r$ | $L_r$ | $f$ |
|---|---|---|---|---|---|---|
| 162.4 uH | 15.59 nF | 162.4 uH | 15.59 nF | 99.98 nF | 25.35 uH | 100 kHz |

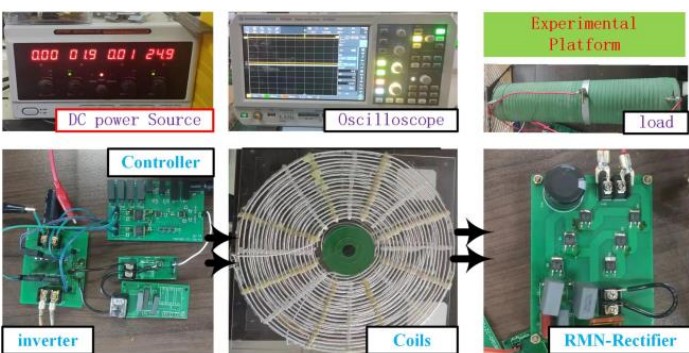

**Figure 13.** The experimental platform for the main circuits part.

### 5.2. Experimental Result

The waveforms of the output voltage $v_p$, output current $i_p$ of the HB inverter, the output voltage $V_o$, and current $I_o$ at different loads are depicted in Figure 14. The output voltage can be adjusted to a constant 36 V by controlling the phase shift angle of the HB inverter. The equivalent duty cycle $D$ for 10 Ω and 25 Ω are 0.47 and 0.26, respectively. Additionally, the zero-phase angle between $v_p$ and $i_p$ is realized as well due to the pure resistive conversion characteristics of the proposed RMN structure. As a result,

nearly no reactive power will be injected into the WPT system, and the power loss can be greatly reduced.

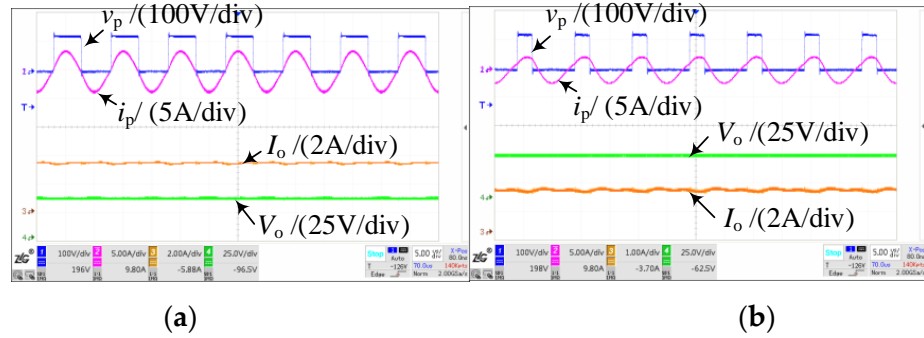

**Figure 14.** The input and output waveforms (**a**) at $R_L$ = 10 ohm and (**b**) at $R_L$ = 25 ohm.

The input voltage $v_{ab}$ and current of RMN at load 25 Ω are drawn in Figure 15. From the waveforms of $v_{ab}$ and $i_s$, the initial phase of $v_{ab}$ and $i_s$ are −1.19 and −1.34, respectively. Therefore, there is almost no phase angle error between them, which proves that almost no additional imaginary impedance is brought during the whole nonlinear conversion process. Compared with the passive matching network, the proposed RMN rectifier has a better application potential.

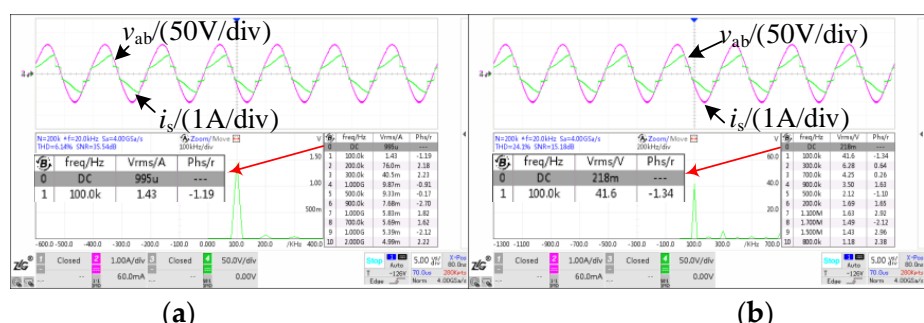

**Figure 15.** The waveforms of RMN at $R_L$ = 25 Ω, (**a**) $i_s$ and (**b**) $v_{ab}$.

The efficiency curve (from the dc source to the load side) against load $R_L$ is shown in Figure 16. The measured transmission efficiency curve for traditional topology without RMN is also given in Figure 16. The efficiency comparison can be divided into two parts to analyze. (1) When the load increases from 10 Ω to 40 Ω, for the topology without the proposed RMN structure, the FB inverter is required to offer enough power for the load. Moreover, the switching loss of power switches dominates the whole system loss. Therefore, the transmission efficiency of the system utilizing the HB inverter with RMN structure is higher than that of the system utilizing the FB inverter without RMN structure. (2) When the load increases from 40 Ω to 50 Ω, the HB inverter can offer enough power to the load for the traditional topology without RMN. Therefore, the HB inverter can be used to replace the FB inverter for the traditional topology without RMN. Therefore, the efficiency difference between the topology with RMN and the topology without RMN will be narrowed down slowly. Lastly, they can reach the same transmission efficiency.

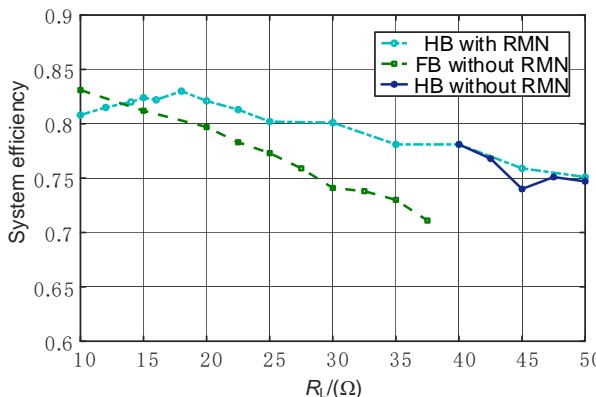

**Figure 16.** The measured efficiency curves.

## 6. Conclusions

In this article, a new topology, by introducing a load compression network before the traditional bridge rectifier, is proposed for the WPT system. The inherent mechanism of efficiency reduction for WPT systems utilizing a phase shift control during a wide load range is studied. Then, poor transmission efficiency for the WPT system utilizing a phase shift control can be improved by replacing the FB inverter with an HB inverter. The insufficient power capability of the HB inverter under a small load value can be solved by the addition of the RMN structure. Therefore, for the WPT system utilizing phase shift control, transmission efficiency can be improved, and constant voltage output can be satisfied in a wide load range. Finally, an experimental platform with a 36 V output voltage is constructed to verify the practical performance of the proposed WPT system. The dc/dc transmission efficiency of the whole WPT system can be kept above 75% even under poor conditions, which meets well with the theoretical analysis results.

**Author Contributions:** Conceptualization, H.Y. and C.W.; methodology, H.Y.; software, T.C.; validation, H.Y. and T.C.; formal analysis, H.Y.; data curation, H.Y.; writing—original draft preparation, T.C.; writing—review and editing, C.W. All authors have read and agreed to the published version of the manuscript.

**Funding:** This research was not funded by any funding.

**Data Availability Statement:** All the data supporting the reported results have been included in this paper.

**Acknowledgments:** We would like to thank the reviewers for their valuable review comments. Some of the revisions have been taken directly from the reviewers' suggestions.

**Conflicts of Interest:** The authors declare no conflict of interest.

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
