# Peer review of "Efficiency Improvement for Wireless Power Transfer System via a Nonlinear Resistance Matching Network"

_electronics, doi:10.3390/electronics12061341_

Round 1

Reviewer 1 Report

A resistance matching network, including an inductor and capacitor, is combined with rectifier to provide reliable and efficient Wireless Power Transfer (WPT). The variation range of the equivalent AC impedance before the rectifier is optimized with help from nonlinearities. Schematic diagrams for the switching loss of power switch are provided. The topology is optimized and the output power against RL under different rectifier parts are represented.

The paper deals with an interesting topic but certain points should be improved in order to become publishable at Electronics. More specifically:

(i) Graphs of the received power with respect to capacitances and inductances should be provided and interpreted.

(ii) How the efficiencies are affected by the nonlinearity?

(iii) Discuss connections with alternative strategies of WPT [1,2] and performance improvement via nonlinear systems [3,4].

[1] Robust wireless power transfer using a nonlinear parity–time-symmetric circuit, Nature, 2017.

[2] On-site wireless power generation, IEEE Transactions on Antennas and Propagation, 2018.

[3] Angular memory of photonic metasurfaces, IEEE Transactions on Antennas and Propagation, 2021.

[4] Enhancement of nonlinearity by a parasubstituted compounds - it's spectroscopic analysis, Materials Today, 2021.

Reviewer 2 Report

The units of voltage and current per DIV in Figures 14 and 15 can be noted in large fonts.  Too smaller fonts in scope are not clear to see. 

This paper presents in good description and It is accepted as it is.  

Reviewer 3 Report

The research is presented in a concise form. A sufficient number of figures, tables and equations are shown. The information is reported in a balanced manner. The article definitely has scientific value. I believe that it would also be interesting for a wide range of persons interested in the subject. The sources are relatively new and objective.

I have only a few recommendations (which do not detract from the good quality of the research):-

- To expand the conclusion. Also to emphasize in separate paragraphs;

- To apply acknowledgements.

I propose that with these minor corrections, the study be published.

Author Response

Point 1:  To expand the conclusion. Also to emphasize in separate paragraphs;

Response 1:  Thanks for your comments. The conclusion has been improved as

In this article, a new topology, by introducing a load compression network before the traditional bridge rectifier, is proposed for the WPT system. The inherent mechanism of efficiency reduction for WPT system utilizing a phase shift control during a wide load range is studied. Then poor transmission efficiency for the WPT system utilizing a phase shift control can be improved by replacing the FB inverter with an HB inverter. The insufficient power capability of the HB inverter under a small load value can be solved by the addition of RMN structure. Therefore, for the WPT system utilizing phase shift control, transmission efficiency can be improved and constant voltage output can be satisfied in a wide load range. Finally, an experimental platform with 36V output voltage is constructed to verify the practical performance of the proposed WPT system. The dc-dc transmission efficiency of whole WPT system can be kept above 75% even under poor conditions, which meets well with the theoretical analysis results.

Point 2:  To apply acknowledgements

Response 2:  Thanks for your comments.

Acknowledgments: We would like to thank the reviewers for their valuable review comments. Some of the revisions have been taken directly from the reviewers’ suggestions.

Round 2

Reviewer 1 Report

Accept in present form